# Protocol for a scoping review of research on abortion in sub-Saharan Africa

**Kenneth Juma, Ramatou Ouedraogo, Meggie Mwoka, Anthony Idowu Ajayi[ID],
Emmy Igonya, Emmanuel Oloche Otukpa[ID]\*, Boniface Ayanbekongshie Ushie**

African Population and Health Research Center, Nairobi, Kenya

\* Otukpa.emmanuel@gmail.com

**Funding:** This research is supported through the funding to APHRC by the Swedish International Development Cooperation Agency (https://www.sida.se/en). The funders had and will not have a role in study design, data collection and analysis,

## Abstract

### Introduction

Unsafe abortion is a leading cause of maternal mortality, and access to safe abortion services remains a public health priority in sub-Saharan Africa (SSA). A considerable amount of abortion research exists in the region; however, the spread of existing evidence is uneven such that some countries have an acute shortage of data with others over-researched. The imbalance reflects the complexities in prioritization among researchers, academics, and funders, and undeniably impedes effective policy and advocacy efforts. This scoping review aims to identify and map the landscape of abortion research in SSA, summarize existing knowledge, and pinpoint significant gaps, both substantive and geographic, requiring further investigation. This review will provide direction for future research, investments, and offer guidance for policy and programming on safe abortion.

### Materials and methods

We utilize the Joanna Briggs Institute's methodology for conducting scoping reviews. We will perform the search for articles in 8 electronic databases (i.e., PubMed, AJOL, Science Direct, SCOPUS, HINARI, Web of Knowledge, CINAHL, and WHO Regional Databases). We will include studies written in English or French language, produced or published between January 1, 2011, and July 31, 2021, and pertain directly to the subject of abortion in SSA. Using a tailored extraction frame, we will extract relevant information from publications that meet the inclusion criteria. Data will be analyzed using descriptive statistics and thematic analysis in response to key review questions.

### Ethics and dissemination

Formal ethical approval is not required, as no primary data will be collected. The findings of this study will be disseminated through peer-reviewed publications and conference presentations.

decision to publish, or preparation of the
manuscript.

**Competing interests:** The authors have declared
that no competing interests exist.

## Introduction

Abortion is common in sub-Saharan Africa (SSA) [1, 2], and because of limited access to safe
options, the majority of women and girls in need of pregnancy termination use unqualified
and clandestine providers, leading to adverse outcomes [3]. Unsafe abortion and the resulting
complications contribute significantly to maternal morbidity and mortality in the region [4].
Region-wide studies have established an increasing incidence of induced and unsafe abortions
[5, 6]. Annually, from 2010–2014, about 15% of all pregnancies in Africa ended in abortion
[7], of which approximately 8.2 million induced abortions occurred, representing a nearly
100% increase from 1990 and 1994 [1]. Besides demographic changes—specifically, an increase
in the number of women of reproductive age [7]—, other reasons for the rise in induced abor-
tions include the growing unmet need for contraception and improvements in technologies
and procedures for terminating pregnancies [5]. Evidence from some low and middle-income
countries (LMICs) points to the use of abortion as a substitute for contraception [8, 9].

Notably, about 75% of all abortions carried out in Africa are unsafe (i.e., pregnancy termi-
nation conducted by providers lacking the necessary skills or in an environment lacking the
minimal medical standards or both) [10]. The risk of maternal death from unsafe abortion
(one in every 150 procedures) is the highest in the world [11, 12]. The latest estimates show
that over 70,000 women die annually from unsafe abortion (the majority being in Africa
(38,000) and South Asia (24,000) [13]. Approximately 6.9 million women in developing
regions suffer unsafe abortion-related complications and morbidities that require treatment,
some with extended hospital stays, and intensive care involving highly skilled, yet scarce,
health providers and enormous financial costs [6, 14].

Arguably, more than 98% of women in SSA live in legally restrictive abortion contexts that
constrain access to safe abortion, with exceptions only in cases of rape or incest or to save a
woman's life [15–17]. Criminalization has remained the primary tool for regulating abortion
in most SSA countries, despite its limited effectiveness in ending or reducing abortions [18,
19]. Rather than limiting abortion rates, criminalization leads to a rise in unsafe abortions,
related complications, and increase maternal morbidity and mortality [11, 20]. Global develop-
ment frameworks long recognized these challenges and proposed bold targets such as the 1994
International Conference on Population and Development (ICPD), where states committed to
ensuring access to quality post-abortion care services, despite existing anti-abortion laws. The
sustainable development goals (SDGs) targets a global average maternal mortality ratio of less
than 70 maternal deaths per 100 000 live births by 2030 [21]. Every country has the mandate to
reduce its national maternal mortality ratio from baseline by two-thirds in that timeframe.
Within this context, and in light of the potential benefits to women's health and well-being of
reforming abortion laws and policies [11], African countries have forged collective commit-
ments to improve access to safe abortion services, including post-abortion care services. Such
commitments encompassed continental frameworks such as the Protocol to the African Char-
ter on Human and Peoples' Rights on the rights of women in Africa (commonly known as the
Maputo Protocol) and the Addis Ababa Declaration on Population and Development in Africa
beyond 2014. Profound gaps remain, however, in the domestication and implementation of
such progressive instruments at sub-regional or national levels.

Strong arguments exist in support of marshaling policy-relevant evidence to help address
the barriers that presently impede the realization of the continental commitments on
improved access to safe and legal abortion. An extensive range of studies has addressed critical
questions concerning abortion in SSA, among others, on its magnitude, causes, and patterns
across particular settings, or levels of access to and utilization, and the quality of abortion and
post-abortion care services. However, the kinds of evidence–in substantive, geographic, or

methodological terms–that are most relevant to promoting policy progress on abortion across SSA's sub-regions and countries may not yet exist. Where such evidence is available, there is the need to interrogate and synthesize it for application. A comprehensive scoping review of the literature is needed to identify and provide directions for addressing such gaps. We aim to conduct such a scoping study to map the landscape of extant research on abortion in SSA, clarify key concepts, summarize existing knowledge and pinpoint gaps, both substantive and geographic, requiring further investigation. Even though there is a protocol published by Coast et al., 2019 focusing on the economics of abortion [22], our proposed study is unique in that it has a broader focus on abortion research targeting SSA. This study will synthesize the existing evidence to facilitate debates and advocacy at various levels in SSA and identify gaps for further inquiry.

## Review questions

i. How has abortion research (e.g., trends in volume, themes, study designs, and African-led studies) evolved in sub-Saharan Africa over the past decade?

ii. What is the geographical landscape of evidence on abortion incidence, the economic burden of unsafe abortion, and the cost and consequences of unsafe abortion to women and girls in sub-Saharan Africa, and what are the key findings?

## Materials and methods

We will apply the Joanna Briggs Institute's approach for conducting a scoping review [23]. The methodology involves a systematic approach to searching, screening, and reporting that encompasses the following stages: (1) identification of the research question (s); (2) identification of relevant databases and studies; (3) selection of studies; (4) data extraction; (5) interpretation, summarization and dissemination of the results.

We will search relevant peer-reviewed, English or French-language articles published between January 1, 2011, and July 31, 2021, without methodological restrictions, in several electronic databases, as well as in general internet sources (Google and Google Scholar).

We focus on articles and reports published between aforementioned dates, because we consider 11 years as a reasonable timeframe to reflect on the extent of research and evidence that is within the realm of 'current' and valid for informing policy processes and debates. Also, focusing on eleven years will yield a manageable number of articles that could be quickly summarized to inform policy processes and discussions on abortion at the continental level and sub-regions.

## Inclusion criteria

We intend to capture all research papers published on abortion in SSA, including those focusing on women, health providers, policymakers, and community members between January 2011 and December 2020. However, to qualify for inclusion, papers have to be:

- Journal articles (peer-reviewed),

- English or French language publications.

We will exclude commentaries, conference abstracts, and posters, working papers, policy briefs, editorials, opinion pieces, and debates. Also, we will exclude technical reports and thesis to avoid double counting. We used Table 1 to present our population, intervention, control, outcomes, timeframe, and settings (PICOTS) [22]. We are interested in studies on women and

**Table 1. PICOTS table summarising our scoping review approach.**

| PICOTS | Micro-level | Meso-level | Macro-level |
|---|---|---|---|
| Population | Girls and women who had abortions or post-abortion care and family members | Communities and health facilities, pharmacies, health providers, where abortion is obtained | Countries, states |
| Interventions | Abortion | | |
| Control | None | | |
| Outcomes | Quantitative, qualitative, mixed methods and review research on abortion incidence and magnitude, economic burden, perceptions of abortion, women who abort, postabortion care, abortion policies and laws, reasons and drivers of abortion, unsafe abortion causes and consequences, quality of post-abortion care, | | |
| Timeframe | January 1, 2011, and July 31, 2021 | | |
| Settings | sub-Saharan Africa | | |

girls, health providers, and community members as well as policymakers on abortion in SSA. We tailored our search and screening approach to mirror the PICOT in Table 1.

## Search strategy for databases

1. We will conduct an initial search using PubMed followed by an analysis of words in titles, abstracts, and indexes to refine search terms in subsequent steps

2. We will use the keywords and controlled vocabulary identified in Step 1 to create Mesh terms. Team members will then test our search terms, and the finalized search strategy will be peer-reviewed by an independent researcher who has previously conducted systematic review studies (using PRESS or a similar method). We will then use the terms to search all included databases. We have presented examples of our search terms in two databases in S1 Appendix. All relevant articles emanating from our search will be imported to Covidence, an online software for managing scoping and systematic reviews [24, 25]. Covidence will automatically remove duplicate articles.

3. The reference list of all identified review studies will be examined to identify other studies not captured through the electronic search. Suitable titles identified through this means will be added to the list of studies for review.

## Databases

We will search the following databases PubMed, HINARI, AJOL, Science Direct, SCOPUS, and CINAHL.

## Search terms

We have included our search term, developed through our preliminary search and reviewed by an independent researcher with experience in systematic reviews, in S1 Appendix. We have searched PubMed and HINARI using these search terms and will report the number of articles found in these databases.

## Study selection

We will use the Covidence software [25]–an online tool that allows reviewers to screen through a plethora of articles simultaneously and for exporting included titles to Excel for analysis. Two researchers will independently assess articles for inclusion by screening the titles, abstracts, and full-texts of studies returned through the search process. Where there are

disagreements between the two independent reviewers on the eligibility of a paper for inclusion, a third reviewer will resolve the conflict.

## Data extraction

We will use a standardized frame (Table 2: Extraction framework) to extract information from the included articles.

**Table 2. Extraction framework.**

| Key domains | Sub-category | Description |
| --- | --- | --- |
| Author | First | Indicate where the lead author is based (e.g., African based In Africa, non-African based in Africa, Africa based outside of Africa and non-African not based in Africa) |
| | Last | |
| Author composition | | Indicate if the paper has a single author or multiple authors |
| Collaboration types | Collaboration analysis | Are the collaborators based in the US/Canada or Europe or multiple locales or Africa? |
| Is there an author from the country of study's focus? | Authorship analysis | Indicate if any of the authors are from the country of study |
| Title | | Indicate study title |
| Language | | Indicate the language of publication |
| Country | | Indicate country (ies) where the study was conducted |
| Study setting | | Specify the location of study sites (eg, district(s) |
| Sub-region | | sub-Saharan Africa, East Africa, West Africa, Central Africa, Southern Africa |
| Type of publication | | Indicate the study type (e.g., primary research, and secondary analysis) |
| Study design | | Indicate if the study adopted a mixed-methods, qualitative, quantitative designs of a review study |
| Aim/objectives | | Describe the stated aim and objectives of the study |
| Focus of study | Intervention study | Indicate whether the study described a problem or examine the effect of an intervention |
| | Describing the problem | |
| Key findings | | Summarize the main results (including an overview of concepts, themes, and types of evidence available), link to the review questions and objectives, and consider the relevance to key groups. |
| Key Limitations | | Indicate the limitations of the study. |
| Publication Year | | |
| Funding | | Describe sources of funding for the included sources of evidence. Describe the role of the funders of the study. Indicate whether the funder (e.g., Africa, European, US, Other north America) |
| Abstract | | |
| Journal | | |
| Journal coverage | | Indicate if the journal is based in the country of study, has and Africa focus or international focus |
| Journal impact factor | | Indicate the journal's impact factor |
| Link | | |
| Type of methodology | | Indicate the study methodology such as qualitative, quantitative or mixed methods, case studies |
| Methodology | Study design | Specify the study design adopted (e.g., cross-sectional design, case study, pre-post study design, longitudinal study) |
| | Population | Describe the characteristics of the target population |
| | Sampling strategy | Describe how sampling was done if applicable |
| | Data collection | Specify the methods of data collection |
| | Outcome measures | Indicate the outcome measures for quantitative studies |
| | Data analysis | Indicate method of data analysis |
| Theme | | Describe the topic (s) addressed in the study |

(*Continued*)

**Table 2.** (Continued)

| Key domains | Sub-category | Description |
|---|---|---|
| | Policy and legal reviews | Studies focusing on laws, policies and abortion guidelines |
| | Abortion care | methods, quality, access, availability, acceptability |
| | Abortion methods | Medical, surgical, traditional |
| | Post-abortion care | Quality, access, availability, acceptability, contraceptive counseling |
| | Sources of abortion | Pharmacies, drug sellers, unregulated clinics, private clinics, traditional healers and public health facilities |
| | Incidence/magnitude of abortion | |
| | Complications/Consequences of abortion | |
| | Abortion stigma | |
| | Costs of unsafe abortion | |
| | Providers | Studies describing provider characteristics (i.e. training, behavior, practice, experiences, values, beliefs etc.) |
| | Patient characteristics | Studies describing women/girls characteristics (i.e. education etc.) |
| | Community perceptions of abortion, women who abort and post-abortion care | |
| | Decision making | |
| | Drivers or facilitator of abortion | |
| | Male engagement | |
| | Maternal mortality | |
| | Causes of abortion | |
| | Conscientious objection | |

## Data synthesis and presentation of results

We will analyze the data using descriptive statistics and thematic analysis, with results organized in tables and charts and presented into themes that reflect the review objectives. Tables will be used to illustrate how abortion research has evolved from January 2011 to December 2020 in terms of volume, themes, study design, African-led papers, and geography. The PRISMA flow chart will be used to summarize our search and studies included and excluded. A summary narrative that synthesizes the information across key themes, including abortion incidence, burden, cost, post-abortion care, and community perception of abortion, will be developed, critically highlighting the advances and gaps in researchers. Supporting figures will also be developed to present the synthesis, with a focus to draw implications for future research.

## Patient and public involvement statement

Patients and the public were not involved in the design of this study. This study is will synthesis publicly available publications, which reported on patients' and public's experiences.

## Ethics and dissemination

Ethical approval is not required, as we are not collecting primary data but rather analyzing already published papers.

The findings of this study will be disseminated through peer-reviewed publications and conferences as well as in relevant stakeholder fora. In case of any amendments to the protocol following its publication, we will provide the date of each amendment, describe the change(s), and report the rationale for the change(s) in future publications arising from this protocol.

## Strengths and limitations of this study

This scoping review will only look at research and publications over 10 years (2011–2020), yet obviously, there are equally important articles preceding that period. We also intend to review data and articles published in English and French only and within sub-Saharan Africa, thus excluding publications in Arabic, Spanish and Portuguese languages. The keywords to be used in the search strategy are broad and may not identify specialized studies in abortion.

## Supporting information

**S1 Checklist. Preferred reporting items for systematic reviews and meta-analyses extension for scoping reviews (PRISMA-ScR) checklist.**
(DOCX)

**S1 Appendix. Database search strategy.**
(DOCX)

## Author Contributions

**Conceptualization:** Ramatou Ouedraogo, Meggie Mwoka, Anthony Idowu Ajayi, Emmy Igonya, Boniface Ayanbekongshie Ushie.

**Data curation:** Anthony Idowu Ajayi, Emmanuel Oloche Otukpa.

**Formal analysis:** Ramatou Ouedraogo, Anthony Idowu Ajayi, Emmanuel Oloche Otukpa.

**Funding acquisition:** Boniface Ayanbekongshie Ushie.

**Investigation:** Kenneth Juma, Ramatou Ouedraogo, Meggie Mwoka, Anthony Idowu Ajayi, Emmy Igonya, Emmanuel Oloche Otukpa, Boniface Ayanbekongshie Ushie.

**Methodology:** Kenneth Juma, Meggie Mwoka, Anthony Idowu Ajayi, Emmy Igonya, Emmanuel Oloche Otukpa.

**Project administration:** Boniface Ayanbekongshie Ushie.

**Supervision:** Ramatou Ouedraogo, Anthony Idowu Ajayi.

**Visualization:** Emmanuel Oloche Otukpa.

**Writing – original draft:** Kenneth Juma, Ramatou Ouedraogo, Meggie Mwoka.

**Writing – review & editing:** Emmanuel Oloche Otukpa.

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
