## [Decision Letter · Decision Letter 0]

16 Jun 2021

PONE-D-21-11694

Protocol for a scoping review of research on abortion in sub-Saharan Africa

PLOS ONE

Dear Dr. Otukpa

Thank you for submitting your manuscript to PLOS ONE. After careful consideration, we feel that it has merit but does not fully meet PLOS ONE’s publication criteria as it currently stands. Therefore, we invite you to submit a revised version of the manuscript that addresses the points raised during the review process.

We look forward to receiving your revised manuscript.

Kind regards,

Ali Rostami

Academic Editor

PLOS ONE

Journal Requirements:

Additional Editor Comments (if provided):

Reviewers' comments:

Reviewer's Responses to Questions

**Comments to the Author**

1. Does the manuscript provide a valid rationale for the proposed study, with clearly identified and justified research questions?

Reviewer #1: Yes

Reviewer #2: Partly

2. Is the protocol technically sound and planned in a manner that will lead to a meaningful outcome and allow testing the stated hypotheses?

Reviewer #1: Yes

Reviewer #2: Yes

3. Is the methodology feasible and described in sufficient detail to allow the work to be replicable?

Reviewer #1: Yes

Reviewer #2: Yes

4. Have the authors described where all data underlying the findings will be made available when the study is complete?

Reviewer #1: Yes

Reviewer #2: Yes

5. Is the manuscript presented in an intelligible fashion and written in standard English?

Reviewer #1: Yes

Reviewer #2: Yes

6. Review Comments to the Author

You may also provide optional suggestions and comments to authors that they might find helpful in planning their study.

Reviewer #1: The protocol is well written. I recommend minor revision before acceptance.

1. throughout the manuscript: references should be modified according to style recommended by PLOS journals.

2. Methods

• Please explain why you selected French-language articles?

• Why you limited your searches up to December 31, 2020. Why you don’t extend this time for articles published articles in 2021?

• Web of science is not a database. Do you mean Web of science core collection?

Reviewer #2: comments can be seen in the attached file, including changing references and updating them, mentioning proper references.

I suggest that this section to be divided into :Research question , Data source and research strategy, citation management, eligibility criteria , title and abstract relevance screening, data characterization, data summary and synthesis.

7. PLOS authors have the option to publish the peer review history of their article (what does this mean?). If published, this will include your full peer review and any attached files.

Reviewer #1: No

Reviewer #2: No

---

## [Author Response · Author response to Decision Letter 0]

23 Jun 2021

All comments have been addressed and responses uploaded

---

## [Editor Report · Decision Letter 1]

5 Jul 2021

Protocol for a scoping review of research on abortion in sub-Saharan Africa

PONE-D-21-11694R1

Dear Dr. Otukpa,

We’re pleased to inform you that your manuscript has been judged scientifically suitable for publication and will be formally accepted for publication once it meets all outstanding technical requirements.

Kind regards,

Ali Rostami

Academic Editor

PLOS ONE
---

## [Editor Report · Acceptance letter]

8 Jul 2021

PONE-D-21-11694R1 

Protocol for a scoping review of research on abortion in sub-Saharan Africa 

Dear Dr. Otukpa:

I'm pleased to inform you that your manuscript has been deemed suitable for publication in PLOS ONE. Congratulations! Your manuscript is now with our production department. 

Kind regards, 

on behalf of

Dr. Ali Rostami 

Academic Editor

PLOS ONE